# Bacterial Ventilator-Associated Pneumonia in COVID-19 Patients: Data from the Second and Third Waves of the Pandemic

**DOI:** 10.3390/jcm11092279

**Published:** 2022-04-19

**Authors:** Alessandro Russo, Vincenzo Olivadese, Enrico Maria Trecarichi, Carlo Torti

**Affiliations:** Infectious and Tropical Disease Unit, Department of Medical and Surgical Sciences, “Magna Graecia” University of Catanzaro, Viale Europa, 88100 Catanzaro, Italy; olivadesevincenzo.95@gmail.com (V.O.); em.trecarichi@unicz.it (E.M.T.); torti@unicz.it (C.T.)

**Keywords:** COVID-19, VAP, ICU, mortality, antimicrobial therapy

## Abstract

During the coronavirus disease 2019 (COVID-19) pandemic, many patients requiring invasive mechanical ventilation were admitted to intensive care units (ICU) for COVID-19-related severe respiratory failure. As a matter of fact, ICU admission and invasive ventilation increased the risk of ventilator-associated pneumonia (VAP), which is associated with high mortality rate and a considerable burden on length of ICU stay and healthcare costs. The objective of this review was to evaluate data about VAP in COVID-19 patients admitted to ICU that developed VAP, including their etiology (limiting to bacteria), clinical characteristics, and outcomes. The analysis was limited to the most recent waves of the epidemic. The main conclusions of this review are the following: (i) *P. aeruginosa, Enterobacterales*, and *S. aureus* are more frequently involved as etiology of VAP; (ii) obesity is an important risk factor for the development of VAP; and (iii) data are still scarce and increasing efforts should be put in place to optimize the clinical management and preventative strategies for this complex and life-threatening disease.

## 1. Introduction

During the coronavirus disease 2019 (COVID-19) pandemic, a huge number of patients have required admission to intensive care units (ICUs) for COVID-19-related severe respiratory failure requiring invasive mechanical ventilation (IMV) [1]. Overall, about 25% of COVID-19 patients require critical care management [2], with a consequent increased risk of developing ventilator-associated pneumonia (VAP) [3,4].

Diagnosis of VAP is challenging for physicians considering the importance of an early assessment of infection, the role of colonization and its interpretation, and the importance of an early appropriate antimicrobial therapy [5]. VAP is associated with a high mortality rate and a considerable burden on length of ICU stay and healthcare costs [6]. Moreover, the significant increase in antimicrobial resistance among bacterial pathogens represents the main challenge for clinicians in ICUs. To date, despite the wide choice of antibiotic therapy, knowledge of the local epidemiology, patient’s risk stratification, and infection control policies (mainly antimicrobial stewardship programs) remain the key elements for the effective management of infections caused by multidrug-resistant (MDR) microorganisms [7].

Considering that the proportion of patients with COVID-19 admitted to ICU who developed VAP has been variably reported [8], and microbiological etiology and outcomes have not well established, the objective of this review is to evaluate data about COVID-19 patients with VAP, including microbiological etiology, clinical characteristics, and outcomes focusing on the “second” and “third” waves of the pandemic.

## 2. Materials and Methods

We conducted research of PubMed (National Library of Medicine 8600 Rockville Pike Bethesda, MD 20894, USA) from January 2021 to December 2021. The keywords used were “VAP”, “mechanical ventilation”, and “COVID-19”, whereby 45 scientific papers were identified. We included all observational, retrospective, and prospective studies. We dismissed all papers concerning non-bacterial-VAP in COVID-19 patients. From the study by Meawed et al. [9], we only reported data regarding bacterial superinfections. No language restrictions were applied in the literature search. Studies involving fewer than 10 patients, case reports, abstracts, and non-peer-reviewed articles were excluded. The selected records were reviewed to verify the inclusion criteria. Finally, 18 articles were included.

The inclusion or exclusion criteria are detailed in the flow diagram (see Figure 1).

## 3. Characteristics of the Included Studies and Study Populations

The design and objectives of the included studies are reported in Table 1.

All these papers were published between January and December 2021, based on data from second and third waves of pandemic. Further, 13 of 18 studies (13/18, 72.2%) had an observational retrospective design, and 12/18 (66.7%) were conducted at a single center, whereas 6/18 (33.3%) were multicenter studies. One was an observational single-center study of prospective data [10], one was a planned ancillary analysis of a multicenter retrospective European cohort [16], and one was a monocentric observational cross-sectional study [9].

All studies were conducted in the European Union, except for one conducted at the Northwestern University Feinberg School of Medicine (Chicago, IL, USA) [10], one at Zagazig University, one at Isolation Hospitals (Zagazig, Egypt) [9], one at Sanz Medical Center, Netanya, Israel [26], and one in a tertiary care center in Mexico City, Mexico [25].

The main objective of these studies was to determine the prevalence and etiology of bacterial superinfections in patients with severe SARS-CoV-2 pneumonia. Only two studies compared COVID-19 patients with non-COVID-19 patients admitted to the ICU who developed VAP [17,19]. Maes et al. showed that COVID-19 patients were more likely to be investigated for VAP and exhibited a higher incidence of microbiologically confirmed VAP (48% compared to 13% in non-COVID-19 group) [17]. In the study of Rouyer et al., COVID-19 patients displayed a significantly higher rate of shock, death in the ICU, VAP recurrence, clinical worsening, positive blood cultures, and polymicrobial cultures compared to non-COVID-19 patients [19]. One study aimed to determinate the impact of SARS-CoV-2 pneumonia on the development of VAP and mortality [16] compared to no-COVID-19 patients; in another study, the incidence of VAP in the study population was evaluated [15] compared to influenza or no viral infection at ICU admission. VAP was associated with an increased 28-day mortality rate and longer durations of IMV and ICU length of stay in COVID-19 patients [16]; compared to influenza and no viral infection, SARS-CoV-2 infection showed no significant impact on the development of VAP and unfavorable outcome (mortality). Conversely, Rouzé et al. [15] showed that the incidence of superinfections of the lower respiratory tract was higher in COVID-19 patients than in influenza or in cases with no viral infections. One other study evaluated the impact of dexamethasone on the incidence of VAP and bloodstream infections (BSI) in COVID-19 patients [13]. In this study, dexamethasone was not associated with an increased incidence of VAP and BSI in patients undergoing IMV, but the data reported in the literature are discordant [27,28].

Based on this evidence, routine antibiotic administration to all COVID-19 patients in the absence of signs of bacterial superinfection should not be recommended. Extensive antibiotic treatment in COVID-19 patients [29] may perturb gut homeostasis, enabling bacterial pathogens to cause pneumonia or other invasive infections [30]. Moreover, inappropriate broad-spectrum antibiotic treatment may increase resistance levels and mortality rates [31]. Pickens et al. reported that early antibiotic treatment should be avoided in over 75% of cases if the gold standard analysis of BAL fluid with multiplex PCR and quantitative culture is appropriately used to identify the etiology of superinfection [10].

A total of 6928 patients with COVID-19 at different stages of disease were analyzed, with a mean of 385 patients per study. The mean age of the population included in these studies was 62.4 years. The percentage of male patients ranged from 60% to 80%. The mean body mass index (BMI) of the patients varied around 28 kg/m^2^, showing the importance of obesity in COVID-19 patients with VAP. In these studies, the definition of chronic disease was not standardized, so we did not report a critical assessment of the role of comorbidities in this population. However, type 2 diabetes and arterial hypertension were very frequent in patients with VAP varying from 16 to 66% and 16.3 to 66.7%, respectively; cardiovascular diseases were reported in 14–40% of patients, while renal disease, particularly chronic renal failure, was reported in 2–21.9% of patients. Finally, chronic obstructive pulmonary disease (COPD) and asthma varied from 3 to 44%. Interestingly, in the analysis of Blonz et al. [11], male sex was associated with a significantly higher occurrence of VAP, but there was no statistically significant relationship between VAP and age, obesity, hypertension, diabetes, chronic respiratory disease, or immunocompromised status. Out of this evidence, the studies included in our analysis did not highlight specific risk factors associated with gender. Thus, a gender-specific analysis may be an important aspect to analyze in future studies.

In the literature, authors have described two different phenotypes of COVID-19 pneumonia according to respiratory tract involvement: type L, characterized by tissue hypoxia and minimal impairment of lung compliance; and type H, which is similar to classic acute respiratory distress syndrome (ARDS), inducing hypoxia and decreased lung compliance [32]. According to Moretti et al. [18], lung compliance was lower in COVID-19 patients who developed VAP compared to those who did not, independent of age, sex, and comorbidities.

## 4. Incidence and Characteristics of VAP

The criteria for the diagnosis of VAP were homogeneous among the studies and were based on criteria adapted from the European Centre for Disease Prevention and Control or the CDC’s National Healthcare Safety Network [5,21,33,34]. Among these studies, the incidence of VAP in critically ill COVID-19 patients was extremely high, varying from 30 to 60%. These data are consistent with those reported in the literature. The incidence of VAP in ICUs varied from 10 to 33% [35] in the pre-COVID era, but the incidence of VAP in COVID-19 patients is reported to be higher than that in non-COVID-19 patients (OR: 3.24), according to a meta-analysis conducted by Ippolito et al. [36].

The median duration of IMV before the development of VAP was 10 (range: 6–17) days. A longer duration of IMV is a well-known risk factor for developing VAP [37], but it can also be a consequence of VAP. However, several studies have demonstrated that the increased risk of developing VAP in COVID-19 patients is not only related to a longer duration of mechanical ventilation [17]. In COVID-19 patients, an important predictor of VAP is the impaired immune cell function [38]. Patients experience a complex dysregulation of their immune system with hyperinflammatory activation and [39] damage to the alveolar membrane, which, although not specific to COVID-19, may also facilitate invasion of bacterial species [35] COVID-19 patients are more likely to present with ARDS, which is an important risk factor for VAP [40]. Prone positioning showed a significantly favorable impact on the clinical outcome, but it may increase the risk of micro-aspiration and VAP [41].

### Key Messages

From a qualitative analysis of data, obesity seems to play a key role in the onset of VAP in critically ill patients with COVID-19.Dysregulation of the immune system, caused by COVID-19, may facilitate VAP onset.Management of VAP, in COVID-19 patients, needs improvement and more data about the relevance of bacterial cultures or isolates from respiratory tract and the role of biomarkers (such as procalcitonin) should be obtained.

## 5. Microbiology

VAP may be caused by a wide spectrum of bacterial pathogens. Common pathogens include both aerobic Gram-negative bacilli, such as *Pseudomonas aeruginosa*, *Escherichia coli*, *Klebsiella pneumoniae*, and *Acinetobacter* spp., and Gram-positive cocci, such as *Staphylococcus aureus* [3]. A summary of the different microorganisms isolated from COVID-19 patients who experienced at least one episode of VAP and reported in the studies included in the present review is presented in Table 2.

The most frequent Gram-positive bacteria were *S. aureus*, accounting for ~30%. *S. aureus* has been previously reported in approximately 70% of the early lower respiratory tract samples from COVID-19 patients [42]. It has been observed that COVID-19 patients, during the first wave of pandemic, were more likely to develop late-onset VAP due to *S. aureus*, including the methicillin-resistant strain, compared to non-COVID-19 patients [43].

Interestingly, our analysis revealed that *P. aeruginosa* and *Klebsiella* spp. were the most frequent Gram-negative bacilli involved in VAP. These species are recognized as very virulent owing to their peculiar phenotypes and virulence genes [44]. The high prevalence of antibiotic resistance and virulence genes in conjunction with a significant relationship between the strains revealed a high pathogenic capacity of the isolated pathotypes of not only *K. pneumoniae*, but also *P. aeruginosa.* Then, several studies demonstrated that Gram-negative bacilli, in particular *P. aeruginosa* and *Enterobacterales*, may cause respiratory infections in ICU settings, exhibiting minimal differences between HAP and VAP in terms of clinical presentation and outcome [45].

The rates of VAP due to MDR pathogens have increased dramatically in ICUs in recent years [46]. In a previous study, 10–50% of the infections were caused by antibiotic-resistant Gram-negative bacteria, and the frequency of MDR pathogens differed depending on the hospital, antibiotic use, and characteristics of ICU patients [38]. Of importance, extended-spectrum β-lactamase-producing *Enterobacterales* (ESBL-E) are reported as the cause of 19–61% of hospital-acquired infections, including VAP [47]. Previous colonization and/or previous antibiotic therapy have been reported to have an important rule on the risk of developing VAP caused by an MDR pathogen [48].

Inappropriate broad-spectrum antibiotic therapy in hospitalized COVID-19 patients may result in a higher incidence of MDR pathogens and higher mortality rate [31]. To date, the association of lung microbiota with poor outcomes [49] remains unclear, and a recent study investigating the lung-tissue microbiota of patients deceased with COVID-19 identified a bacterial community enriched with *Acinetobacter* spp. [50] (including carbapenem-resistant *A. baumannii*) [51]. The microbial richness was not different between COVID-19 and non-COVID-19 patients, but significant microbial diversity has been demonstrated with less low respiratory tract commensal bacteria and more opportunistic pathogens, such as *Pseudomonas* spp., *Enterobacterales*, and *Acinetobacter* spp. [52].

### Key Message

Even though we need to better understand the local epidemiology of MDR pathogens, *P. aeruginosa*, *Enterobacterales* spp., and *S. aureus* are frequently involved in VAP and should be taken into account for empirical antibiotic therapy.

## 6. Impact of Specific COVID-19 Therapy on VAP

In 2020, the first IDSA Guidelines on the Management and Treatment of COVID-19 were released [53]. An important consensus was obtained regarding the management and treatment of COVID-19 patients, with a remarkable impact on the outcome of hospitalized patients.

Of interest, among those authorized for the treatment of COVID-19, some drugs (e.g., corticosteroids or tocilizumab) impact the immune system and may facilitate the onset of superinfections. Regarding the studies included in this review, Gragueb-Chatti et al. [13] focused on the relationship between dexamethasone use and the risk of VAP and BSI. VAP occurred in 63% of patients treated with dexamethasone, but this incidence was not higher than that in the control group. VAP occurred earlier and involved less non-fermenting Gram-negative bacteria, but rather *Enterobacterales*.

Treatment with dexamethasone was associated with more ventilator-free days at day 28, a shorter duration of IMV, and reduced ICU length of stay [27]. Corticosteroids cause immunosuppression mainly by sequestration of CD4+ T-lymphocytes in the reticuloendothelial system and by inhibiting the transcription of cytokines. Then, the prolonged use could aggravate the risk of superinfections, including VAP. Regarding the microbiology of VAP, Gram-negative bacteria (particularly *Enterobacterales* and non-fermenting Gram-negative bacilli) were commonly isolated during the first episode of VAP: *Enterobacterales* were the most frequent etiology in patients treated with dexamethasone, whereas non-fermenting Gram-negative bacilli were more frequent in the control group, although no statistically significant difference was observed between the two groups [13]. VAP recurrence was documented in 37% of the patients, 42% of whom were treated with dexamethasone. The same pathogen was responsible for recurrence in 68% of patients; *Enterobacterales* and *P. aeruginosa* were more frequently associated with relapse [13].

Tocilizumab is a recombinant humanized monoclonal antibody developed against soluble and membrane-bound isoforms of IL-6 receptors. This mechanism is associated with a prolonged immunosuppressive status that could be an important risk factors for superinfections in patients treated with tocilizumab for severe COVID-19. It has been recommended by current guidelines as a treatment for severe ARDS caused by the cytokine storm syndrome [22]. Despite tocilizumab’s immunosuppressive effect, Taramasso et al. [54] did not find a statistical difference in infectious complications between patients treated with tocilizumab and the control group. Therefore, clinical presentations did not differ in the two groups, except for CRP levels, which were reduced at the time of infection onset in patients treated with tocilizumab.

Baricitinib, an orally administered selective inhibitor of Janus kinase (JAK) 1 and 2, should only be administered in combination with dexamethasone or other corticosteroids in patients with increasing oxygen needs and systemic inflammation [52]. Baricitinib can modulate downstream inflammatory responses via JAK1/JAK2 inhibition and has exhibited dose-dependent inhibition of IL-6-induced STAT3 phosphorylation. It has been reported that patients receiving baricitinib plus remdesivir had lower incidence of adverse events, including secondary infections [55]. Additionally, the use of baricitinib associated with corticosteroids has not been associated with an increase in infections, including serious infections or opportunistic infections, in hospitalized patients [56]. However, we did not find any data about the incidence of VAP in patients treated with baricitinib.

### Key Messages

VAP occurrence seems not to be related to immunomodulatory treatments used for COVID-19; however, the use of corticosteroids and tocilizumab may alter the clinical presentation of secondary pulmonary infections.Data about the incidence of VAP in patients treated with JAK-inhibitors, including baricitinib, are needed.Targeted use of antimicrobial therapy is recommended to avoid increase of antimicrobial resistance.Fast microbiology techniques can help physicians for better management of VAP in COVID-19 patients.

## 7. Discussion

Limited information exists about frequency and etiology of pulmonary co-infections and superinfections in patients with COVID-19. VAP is an important complication of patients with COVID-19 requiring IMV, with a negative impact on survival. Several reports revealed that VAP can occur in up to 20–40% of patients admitted to the ICU [57,58], with a variability usually attributable to differences in the clinical setting or the characteristics of patients admitted to the ICU [59]. In regard to COVID-19 patients, no univocal data are available on the incidence of bacterial infections. For instance, a study conducted in China reported that only 13.9% of patients admitted to ICUs for critical COVID-19 pneumonia showed secondary bacterial infections [60].

Data reported in this review are in line with a meta-analysis conducted by Ippolito et al. [36]: nearly half of COVID-19 patients admitted to the ICU may develop VAP, with a pooled estimate of mortality of 42.7% for COVID-19 patients who developed VAP [36]. A clear association between clinical comorbidities and the incidence of VAP was not definitively assessed. Therefore, it appears that several features associated with severe COVID-19, such as ARDS, may predispose patients to VAP, including pulmonary tissue damage, alterations in the lung microbiome, and impairment in lung compliance. Patients with COVID-19 admitted to the ICU are generally severely hypoxemic, displaying both parenchymal and microvascular lung damage [14]. Prolonged IMV, prone positioning, and immunosuppressive and/or immunomodulatory therapies may increase the risk of developing VAP [61,62].

Some issues may also reduce the adherence to infection control protocols and infection prevention bundles. During the waves of pandemic, the ICUs may have been overcrowded with a high risk of inadequate staffing and consequent cross-contamination [63]. Healthcare workers might have some issues with the enforcement of the standards of infection control, focusing on self-protection and feeling a great fear of contagion [64].

Regarding microbiological findings, *Enterobacterales*, among the Gram-negative bacteria, and *Staphylococcus aureus*, among Gram-positive bacteria, were the most frequent bacterial species isolated from cultures collected in patients with suspected VAP. Nevertheless, the distribution of pathogens associated with VAP varies in different countries; therefore, empiric antibiotic treatment should be guided by local microbiological epidemiology and antibiotic resistance data [65]. MDR bacteria and inappropriate initial antibiotic treatment are well-known risk factors for mortality in patients with VAP [12]. Currently, there is no accordance either for or against empiric broad-spectrum antimicrobial therapy in the absence of another indication [53]. Nonetheless, it has been reported that high rates of COVID-19 patients had received broad-spectrum antibiotic treatment before ICU admission [66]. However, the pros and cons of empiric antimicrobial agents in severe COVID-19 patients have not been evaluated in clinical trials.

In addition, the assessment of risk factors for MDR pathogens includes individual patient risk profiles and previously available microbiological data about infection or colonization [7,29]. The clinical deterioration caused by severe COVID-19 could be mistaken for an incoming superinfection and justify empiric antibiotic treatment. Nevertheless, it is now well known that antibiotic treatment, particularly the use of azithromycin, is not associated with better outcomes in hospitalized COVID-19 patients [67]. Only patients with clinical or radiological suspicion of bacterial coinfection should receive antibiotics, with no recommendation for routine use [68].

After almost 2 years of pandemic, our approach to treating the disease has improved, and a new standard of care is now available. SARS-CoV-2 infection promotes an intense cytokine storm, which can dysregulate the innate immune system and facilitate bacterial infections [69]. The use of corticosteroids and immunomodulatory therapies, such as anakinra or tocilizumab, shows promising benefits in patients with severe COVID-19 [70,71]. However, limited data on the impact of these therapies on bacterial coinfections are available. Notably, since these therapies are available for a short time, most of the studies included in this review showed an important bias, considering that immunomodulant therapies were not routinely administered with substantia differences about dose and time of administration. A single-center study conducted in Nijmegen (the Netherlands), showed that PCT and CRP levels were suppressed by dexamethasone treatment and that, after completion of the dexamethasone course, a clear inflammatory rebound effect was observed for both these biomarkers, particularly for CRP. In addition, in patients treated with both dexamethasone and tocilizumab, PCT levels increased following discontinuation of dexamethasone therapy. Furthermore, combined treatment with dexamethasone and tocilizumab appeared to suppress CRP levels, resulting in considerably reduced efficacy in detecting secondary infections [72]. These new findings highlight how the diagnosis and treatment of bacterial coinfections in hospitalized COVID-19 patients remain a challenge for clinicians.

Considering the factors mentioned above, VAP in COVID-19 patients should be considered a challenging complication in terms of diagnosis and management. There are important unmet needs that should be investigated: risk factors (i.e., previous antibiotic therapies and/or immunosuppressive treatment for COVID-19), incidence and prognosis of MDR bacterial infections, effects of antibiotic stewardship, and infection control strategies on the incidence of VAP and outcomes of patients.

## 8. Conclusions

In this review, we report a summary of recent evidence in terms of epidemiology, clinical features, and management of VAP in COVID-19 patients, focusing on the second and third waves of the pandemic. Indeed, the limited sample size of the included studies did not enable us to draw any definitive conclusions. Moreover, the studies available are heterogeneous in terms of microbiological findings, severity of patients’ clinical conditions, antimicrobial therapies, or COVID-19 management. From this review, we can conclude that VAP in COVID-19 patients is peculiar and needs more studies to improve clinical management and elaborate specific guidelines to manage this condition [7,73,74,75,76,77,78,79,80,81,82,83,84,85,86,87].

### Key Messages

Regarding COVID-19 patients, no univocal data are available on the incidence of bacterial infections in VAP.Antimicrobial stewardship programs should be carefully implemented in COVID-19 units, especially the ICU.The assessment of risk factors for MDR pathogens includes individual patient risk profiles and previously available microbiological data about infection or colonization that should be carefully evaluated in every patient.Data about the new licensed antibiotics for the treatment of VAP caused by MDR pathogens should be obtained.

## Figures and Tables

**Figure 1 jcm-11-02279-f001:**
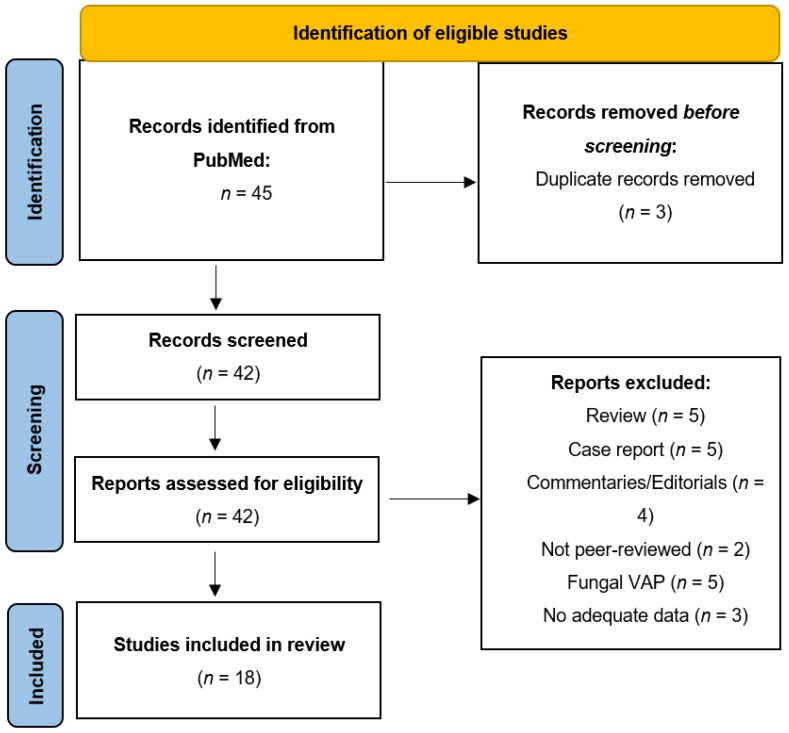
Flow diagram for records identification and screening. Ventilator-associated pneumonia (VAP). Legend: Ventilator-associated pneumonia (VAP).

**Table 1 jcm-11-02279-t001:** Design and objectives of the studies.

Authors	Design (Country)	Objectives
Pickens CO. et al. [10]	Observational single-center study (Illinois, USA)	Prevalence and etiology of bacterial superinfection at the time of initial intubation Incidence and etiology of bacterial VAP
Blonz G. et al. [11]	Multicenter retrospective study (France)	Epidemiological and microbiological description of VAP
Grasselli G. et al. [12]	Multicenter retrospective analysis of prospectively collected data (Italy)	Association with characteristics of critically ill patients with COVID-19 and hospital-acquired infections Association of hospital-acquired infections with clinical outcomes
Gragueb-Chatti I. et al. [13]	Multicenter observational retrospective study (France)	Incidence of VAP and BSI according to the use of dexamethasone Ventilator-free days (VFD) at day 28 and day 60 ICU and duration of hospital stay and mortality.
Giacobbe D.R. et al. [14]	Multicenter observational retrospective study (Italy)	Incidence rate of VAP 30-day case fatality of VAP 30-day case fatality of BALF-positive VAP
Rouzè A. et al. [15]	Multicenter retrospective European cohort performed in 36 ICUs (France, Spain, France, Portugal, and Ireland)	Relationship between SARS-CoV-2 pneumonia, compared to influenza pneumonia or no viral infection, and the incidence of VA-LRTI.
Nseir S. et al. [16]	Planned ancillary analysis of a multicenter retrospective European cohort.	28-day all-cause mortality Duration of mechanical ventilation ICU length of stay censored at 28 days
Maes M. et al. [17]	Retrospective observational study (UK)	Incidence of VAP Bacterial lung microbiome composition of ventilated COVID-19 and non-COVID-19 patients
Moretti M. et al. [18]	Retrospective monocentric observational study (Belgium)	Predictors of VAP in a cohort of mechanically ventilated COVID-19 patients
Rouyer M. et al. [19]	Monocentric retrospective cohort (France)	Death in ICU Death at the end of antibiotic treatment, in-hospital death Duration of intubation, length of hospital stay, length of antibiotic treatment MDR bacterial acquisition Clinical improvement at days 3 and 7 of antibiotic treatment
Meawed TE et al. [9]	Cross-sectional study (Egypt)	Epidemiology of bacterial and fungal VAP in COVID-19 patients.
Garcia-Vidal C. et al. [20]	Retrospective observational cohort study (Spain)	Epidemiology and outcomes of co-infections and superinfections occurring in COVID-19.
Richards O. et al. [21]	Retrospective single-center observational study (UK)	Comparison between PCT and other common biomarkers in revealing or predicting microbiologically proven secondary bacterial infections in an ICU COVID-19 patient.
Taramasso L. et al. [22]	Single-center retrospective case series (Italy)	Clinical presentation of infections in critically ill COVID-19 patients treated with tocilizumab. Comparison of laboratory parameters in patients treated with tocilizumab and not.
Karolyi M. et al. [23]	Retrospective observational study (Austria)	Analyze the spectrum of pathogens detected with BioFire ^®^ Pneumonia Panel from tracheal aspirate or BALF in COVID-19 patients in ICU.
Suarez-de-la-Rica A. et al. [24]	Single-center retrospective observational study (Spain)	Rate of infections in in COVID-19 critically ill patients Analyze risk factors for infections Analyze risk factors for mortality
Martinez-Guerra BA. et al. [25]	Single-center prospective cohort study (Mexico)	Describe empirical antimicrobial prescription Prevalence of HAI Susceptibility antimicrobial patterns
Cohen R et al. [26]	Retrospective observational study (Israel)	Assess the rates and characteristics of pulmonary infections Valuate outcomes of ventilated patients

Legend: Ventilator-associated pneumonia (VAP); bronchoalveolar lavage fluid (BALF); ventilator associated–low respiratory tract infections (VA-LRTI); intensive care unit (ICU); procalcitonin (PCT).

**Table 2 jcm-11-02279-t002:** Etiology of VAP in published studies.

Authors	Gram-Negative	Gram-Positive	MDR
Pickens CO. et al. [10]	*H. influenzae* 7%, *Stomatococcus* spp. 7%, *K. oxytoca* 4%, *M. catarrahalis* 4%, *P. mirabilis* 4%, *Serratia marcescens* 4%, *Stenotrophomonas maltophilia* 4%	MSSA 39%, *Streptococcus* spp. 44%, *Enterococcus* 4%,	MRSA 7%
Blonz G. et al. [11]	*Enterobacteria* 49.8% *Pseudomonas aeruginosa* 15.1%, (*Stenotrophomonas maltophilia*, *Haemophilus*, *Acinetobacter baumannii*, other *Pseudomonas*, etc.) 10.2%	*Staphylococcus aureus* 13.7%, (*Streptococcus pneumoniae*, *Streptococcus agalactiae*, *Corynebacteria*, *Enterococcus faecium*, etc.) 5.9%, *Enterococcus faecalis* 5.4%	MRSA 1.5% *Enterobacterales* 3GC-resistant 52.5%
Grasselli G. et al. [12]	*P. aeruginosa* 21% *Enterobacterales* 14% *Klebsiella* spp. 11% *A. baumannii* 2%	*S. aureus* 28% *Enterococcus* spp. 5% *S. pneumoniae* 1%	MRSA 51% *P. aeruginosa* 12% *Enterobacterales* 11% *Enterococcus* spp. 11%
Gragueb-Chatti I. et al. [13]	*Enterobacteriaceae* 64% *K. pneumoniae* 20% *K. aerogenes* 22% *K. variicola* 4% *K. oxytoca* 4% *Enterobacter cloacae* 13% Non-fermenting GNB 32% including P. *aeruginosa* 81% *S. maltophilia* 11% *Acinetobacter* spp. 7%	MSSA 58% *Enterococcus* 19% *Corynebacterium* 5%	MRSA (7%)
Giacobbe D.R. et al. [14]	*P. aeruginosa* 36% *K. pneumoniae* 19%	*S. aureus* 23%	MRSA 10% CR Gram-negative bacteria 32%
Rouzè A. et al. [15]	*P. aeruginosa* 22.3% *Enterobacter* spp. 18.8% *Klebsiella* spp. 11.5% *E. coli* 8.4% *A. baumannii* 7.3% S. maltophilia 3.5% *S. marcescens* 3.1% *C. freundii* 2.1% *P. mirabilis* 1.7% *H. influenza* 1% *M. morganii* 1%	MSSA 9.4% *Enterococcus* spp. 3.1% *S. pneumoniae* 2.8% *Streptococcus* spp. 1.4%	MDR bacteria 23.3% MRSA 9.4%
Nseir S. et al. [16]	*P. aeruginosa* 24.9% *Enterobacter* 18% *Klebsiella* spp. 12.7% *E. coli* 9.2% *A. baumannii* 4.4% *S. maltophilia* 2% *S. marcescens* 4.4% *Citrobacter freundii* 2.9% *P. mirabilis* 2.4% *H. influenza* 1.5% *M. morganii* 1%	*Enterococcus* 3.4% *S. pneumoniae* 3.4% *Streptococcus* spp. 0.5%	MDR 20.7%, with 2.9% of MRSA
Maes M. et al. [17]	*Klebsiella* spp. *P. aeruginosa* *E. coli* *S. maltophilia*	*S. aureus* *E. faecium* CoN *Staphylococci*	not analyzed
Moretti M. et al. [18]	*K. pneumoniae* 25.9% *K. oxytoca* 11.11% *K. aerogenes* 7.4% *P. aeruginosa* 18.5% *Enterobacter* spp. 11.11% *P. mirabilis* 3.7% *S. marcescens* 3.7% *S. maltophilia* 3.7%	*S. aureus* 7.4%	MDR 66.67% including ESBL *Klebsiella* spp. (29%); XDR 4.76% (1 *P. aeruginosa* VIM-producer)
Rouyer M. et al. [19]	*Enterobacterales* 55% *P. aeruginosa* 19%. Other Gram-negative bacteria 7%.	Gram-positive bacteria 29%	MDR 27%
Meawed TE et al. [9]	*K. pneumoniae* 41.1% *A. baumannii* 27.4% *P. aeruginosa* 20.8% *E. coli* 1.5%	Not specified	PDR *K. pneumoniae* 41.1% XDR *A. baumannii* 27.4% ESBL *P. aeruginosa* 20.8% ESBL *E. coli* 9.1% MRSA 9.1%
Garcia-Vidal C. et al. [20]	*P. aeruginosa* 27.3% *S. maltophilia* 18.2% *K. pneumoniae* 9% *S. marcescens* 9%	*S. aureus* 36.5%	MDR Gram-negative bacteria were isolated in 7 patients: 3 were *P. aeruginosa*, 2 ESBL *E. coli*, 2 ESBL *K. pneumoniae*
Richards O. et al. [21]	Not analyzed	Not analyzed	Not analyzed
Taramasso L. et al. [22]	*P. aeruginosa* *K. pneumoniae* *S. maltophilia* *P. mirabilis* *H. influenzae* *S. marcescens* *E. aerogenes*	*S. aureus* *S. pneumoniae*	Not specified
Karolyi M. et al. [23]	*K. pneumoniae* *H. influenzae* *E. coli* *P. aeruginosa* *S. marcescens* *K. oxytoca* *A. baumannii* *E. cloacae* *Proteus* spp.	*S. aureus* *S. pneumoniae* *S. agalactiae*	Not detected
Suarez-de-la-Rica A. et al. [24]	*Klebsiella* spp. 25.7% *P. aeruginosa* 31.4% *E. coli* 11.4% *Serratia* spp. 5.7%	*S. aureus* (22.8%)	MDR bacteria were detected in 15.9% patients: *Enterobacterales* ESBL; VIM-producing *K. pneumoniae*; MRSA.
Martinez-Guerra BA. et al. [25]	*Enterobacter complex* 42% *P. aeruginosa* 14.5% *Klebsiella* spp. 13% *S. maltophilia* 8.7%	Not specified	AmpC producers 37.7% ESBL producers 8.7% CRE 4.3%
Cohen R et al. [26]	*P. aeruginosa* 41.9% *K. pneumoniae* 22.5% *H. influenzae* 12.9% *E. cloacae* 9.6% *K. aerogenes* 8% *S. marcescens* 6.4% *E. coli* 3.2% *Proteus* spp. 3.2% *M. catarrhalis* 3.2% *A. baumannii* 1.6%	*S. aureus* 37% *S. pneumoniae* 6.4% *S. agalactiae* 4.8%	MRSA CTX-M gene

Legend: multidrug resistant (MDR); methicillin-susceptible *Staphylococcus aureus* (MSSA); methicillin-resistant *Staphylococcus aureus* (MRSA); non-fermenting Gram-negative bacteria (GNB); coagulase-negative staphylococci (CoNS); pandrug resistant (PDR); extensively drug resistant (XDR); extended-spectrum beta-lactamases (ESBL); Verona integron-encoded metallo-β-lactamase (VIM); Carbapenem resistant *Enterobacterales* (CRE).

## Data Availability

On request data area available at a.russo@unicz.it.

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
