# Peer review of "Bacterial Ventilator-Associated Pneumonia in COVID-19 Patients: Data from the Second and Third Waves of the Pandemic"

_jcm, 2022, doi:10.3390/jcm11092279_

Round 1

Reviewer 1 Report

I think this article warrants publication. It is well researched and written and exposes a number of different things in this field, which is not well understood, but nevertheless a large number of guidelines on it have been written. i like the part about limiting anti-microbials and i like how studies were whittled down to the relevant ones. there are no grammatical errors. it is good that they have highlighted that local databases and local epidemiology will suggest how various ICU's approach this

i have 2 specific comments

  1. i am not sure this is a systematic review and if it is, it should be registered on PROSPERO and bias of the studies should be assessed using the QUADAS tools. Perhaps call it a narrative review?
  2. they suggest further research is required, but maybe a list of bullet points to suggest what is actually needed and how they suggest that such research is started/funded

when the above is done, the whole thing is suitable for publication

Author Response

I think this article warrants publication. It is well researched and written and exposes a number of different things in this field, which is not well understood, but nevertheless a large number of guidelines on it have been written. i like the part about limiting anti-microbials and i like how studies were whittled down to the relevant ones. there are no grammatical errors. it is good that they have highlighted that local databases and local epidemiology will suggest how various ICU's approach this

i have 2 specific comments

  1. i am not sure this is a systematic review and if it is, it should be registered on PROSPERO and bias of the studies should be assessed using the QUADAS tools. Perhaps call it a narrative review?

R: Dear reviewer, we are grateful for your important comments. According with your suggestions we consider this manuscript as a narrative review and modified it.

  1. they suggest further research is required, but maybe a list of bullet points to suggest what is actually needed and how they suggest that such research is started/funded

R: we introduced at the end of discussion section a paragraph about future directions and a list of bullet points.

when the above is done, the whole thing is suitable for publication.

Reviewer 2 Report

Dear Authors

Thank you very much for your manuscript submission. Your study is interesting; however, a Major Revision is needed.

  1. Gram-positive/Gram-negative is correct.
  2. The number of tables should be rechecked.
  3. There is no Table 1 in your manuscript.
  4. 12/18=66.7%
  5. One not 1
  6. Please compare the male and female patients' clinical demonstrations predisposing factors, etc with each other through effective tables.
  7. You have mentioned "Interestingly, our analysis revealed that P. aeruginosa and Klebsiella spp. were the most
     frequent gram-negative bacilli involved in VAP.
    it is recommended to explain regarding these bacteria within the main text. In this regard, please read and add the following paper to the References section of the manuscript:

    Virulence factors, antibiotic resistance patterns, and molecular types of clinical isolates of Klebsiella Pneumoniae. Expert Rev Anti Infect Ther. 2022 Mar;20(3):463-472. doi: 10.1080/14787210.2022.1990040. Epub 2021 Oct 28. PMID: 34612762.

  8. You have mentioned the key drugs  of dexamethasone, remdesivir etc. in the manuscript. Please mention the related mechanisms of these drugs and discuss about their roles in the occurrence of the bacterial infections. The use of figures can be very effective.
  9. Please discuss about the age ranges of the female and male patients in this regard.
  10. Discussion and Conclusion sections should be revised.

Author Response

Dear Authors

Thank you very much for your manuscript submission. Your study is interesting; however, a Major Revision is needed.

R: Dear reviewer, thank for you important comments. We modified the manuscript accordingly.

  1. Gram-positive/Gram-negative is correct.
  2. The number of tables should be rechecked.
  3. There is no Table 1 in your manuscript.

R: We modified text. Table 1 and Table 2 are at the end of the text, after conclusions.

  1. 12/18=66.7%
  2. One not 1

R: we corrected it

  1. Please compare the male and female patients' clinical demonstrations predisposing factors, etc with each other through effective tables.

R: about this comment we need to specify that in the analyzed studies the predisposing factors in female and male were not specifically addressed. Then, we cannot report a valuable analysis about the different risk factors for VAP associated with sex. We reported a sentence about the need of gender-specific studies.

  1. You have mentioned "Interestingly, our analysis revealed that P. aeruginosa and Klebsiella spp. were the most
     frequent gram-negative bacilli involved in VAP.
    it is recommended to explain regarding these bacteria within the main text. In this regard, please read and add the following paper to the References section of the manuscript:

    Virulence factors, antibiotic resistance patterns, and molecular types of clinical isolates of Klebsiella Pneumoniae. Expert Rev Anti Infect Ther. 2022 Mar;20(3):463-472. doi: 10.1080/14787210.2022.1990040. Epub 2021 Oct 28. PMID: 34612762.

R: we cited this manuscript (reference n°40) and briefly discussed these characteristics in discussion section.

  1. You have mentioned the key drugs of dexamethasone, remdesivir etc. in the manuscript. Please mention the related mechanisms of these drugs and discuss about their roles in the occurrence of the bacterial infections. The use of figures can be very effective.

R: the analysis of mechanisms of drugs used for COVID-19 is out of the scope of this manuscript. However, we think that your suggestion is very important, then we briefly reported mechanisms of these drugs related to the increased risk of bacterial superinfections and VAP.

  1. Please discuss about the age ranges of the female and male patients in this regard.

R: also about this comment we need to specify that in the analyzed studies the age ranges in female and male were not specifically addressed. Then, we cannot report a valuable analysis about this aspect in our study population.

  1. Discussion and Conclusion sections should be revised.

R: we modified these sections according with reviewers’ comments, including a key messages section at the end of conclusions section.

Round 2

Reviewer 2 Report

Dear Authors

Thank you for your effective revisions. However, you have forgotten to revise  the captions of table 1 (on page 10) and table 2 (on page 13).

Please do revise the number of tables in captions, too (table 2 will be table 1 and table 3 will  be table 2).

Author Response

Dear reviewer, 

we decided not to change the order of the tables, since Table 1 reports the methodological characteristics of the studies and in our opinion it must be reported first, as it is also in the text. We hope you can understand this decision.